# Synthesis, Characterization, Biological Evaluation, and In Silico Studies of Imidazolium-, Pyridinium-, and Ammonium-Based Ionic Liquids Containing *n*-Butyl Side Chains

**DOI:** 10.3390/molecules27196650

**Published:** 2022-10-06

**Authors:** Rabia Hassan, Farzana Nazir, Mah Roosh, Arshemah Qaisar, Uzma Habib, Abdulrahim A. Sajini, Mudassir Iqbal

**Affiliations:** 1Department of Chemistry, School of Natural Sciences (SNS), National University of Sciences & Technology (NUST), H-12, Islamabad 44000, Pakistan; 2Department of Computational Sciences, Research Center for Modeling and Simulation (RCMS), National University of Sciences & Technology (NUST), H-12, Islamabad 44000, Pakistan; 3Healthcare Engineering Innovation Center (HEIC), Department of Biomedical Engineering, Khalifa University, Abu Dhabi P.O. Box 127788, United Arab Emirates

**Keywords:** imidazolium, pyridinium, quaternary ammonium, ionic liquids, antibacterial activity

## Abstract

Ionic liquids (ILs) have emerged as active pharmaceutical ingredients because of their excellent antibacterial and biological activities. Herein, we used the green-chemistry-synthesis procedure, also known as the metathesis method, to develop three series of ionic liquids using 1-methyl-3-butyl imidazolium, butyl pyridinium, and diethyldibutylammonium as cations, and bromide (Br^−^), methanesulfonate (CH_3_SO_3_^−^), bis(trifluoromethanesulfonyl)imide (NTf_2_^−^), dichloroacetate (CHCl_2_CO_2_^−^), tetrafluoroborate (BF_4_^−^), and hydrogen sulfate (HSO_4_^−^) as anions. Spectroscopic methods were used to validate the structures of the lab-synthesized ILs. We performed an agar well diffusion assay by using pathogenic bacteria that cause various infections (*Escherichia coli*; *Enterobacter aerogenes*; *Klebsiella pneumoniae*; *Proteus vulgaris*; *Pseudomonas aeruginosa*; *Streptococcus pneumoniae*; *Streptococcus pyogenes*) to scrutinize the in vitro antibacterial activity of the ILs. It was established that the nature and unique combination of the cations and anions were responsible for the antibacterial activity of the ILs. Among the tested ionic liquids, the imidazolium cation and NTf_2_^−^ and HSO_4_^−^ anions exhibited the highest antibacterial activity. The antibacterial potential was further investigated by in silico studies, and it was observed that bis(trifluoromethanesulfonyl)imide (NTf_2_^−^) containing imidazolium and pyridinium ionic liquids showed the maximum inhibition against the targeted bacterial strains and could be utilized in antibiotics. These antibacterial activities float the ILs as a promising alternative to the existing antibiotics and antiseptics.

## 1. Introduction

The development of robust broad-spectrum antibacterial drugs to treat common skin and soft tissue infections (SSTIs) is a major challenge in the healthcare community. Various chemical synthetic products have been reported to combat pathogenic bacteria [1]. The continuous emergence of antibiotic resistance makes it so that the development of new antibiotics requires time [2]. SSTIs affect workers, families, companies, and countries by compromising human health. The different species of pathogenic bacteria that cause SSTIs are: *Escherichia coli* (*E. coli*); *Klebsiella aerogene/Enterobacter aerogenes* (*K. aerogene*); *Klebsiella pneumoniae* (*K. pneumoniae*); *Proteus vulgaris* (*P. vulgaris*); *Pseudomonas aeruginosa* (*P. aeruginosa*), namely, *Streptococcus pyogenes* (*S. pyogenes*) and *Streptococcus pneumoniae* (*S. pneumoniae*). Gram-positive cocci cause one out of three skin-infection wounds, while Gram-negative bacteria, such as *E. coli* and *K. pneumoniae*, are under observation and alert by the World Health Organization because of their increasing resistance against antibiotics [3]. Ionic liquids in the pharmaceutical and health community have been the center of attention over the past two decades, as they are “green solvents” and are highly explored for drug formulations.

Organic salts that are characterized by melting points below 100 °C are named ionic liquids (ILs). Ionic liquids (ILs) are good examples of neoteric solvents that are contemplated as process solvents, and they find applications as catalysts, plasticizers, coating, biomass dissolution, and battery electrolytes [4]. ILs are a widely accepted choice in place of biphasic processes and organic solvents [5]. These molten salts comprise a unique combination of organic cations and organic or inorganic anions, owning properties such as low melting points [6], high viscosity and density [7], solubility [8], high thermal stability [9], and negligible vapor pressure [10].

Ionic liquid comprises organic cations, such as imidazolium (Im), pyridinium (Py), ammonium (Am), etc., which are attached to various alkyl chains, and anions, such as nitrate, halogenides, phosphor, tetrafluoroborate, trifluoromethanesulfonate, bistriflimide (Tf2N^−^), and hexafluorophosphate (PF6^−^). Various combinations of these anions and cations bring different properties to ILs, such as miscibility and hydrophobicity/hydrophilicity, whereas the alkyl chain length can be varied to fine-tune the desired hydrophobicity [11]. Designed ILs for specific applications are known as task-specific ILs [12].

ILs that contain fluoride in water are notorious for their corrosive properties. ILs are toxic to bacteria but are environmentally friendly [13]. This interesting property makes this novel class a very interesting candidate for their antibacterial [3,14], antibiofilm [15], and wound-healing potentials [16].

Imidazolium- and pyridinium-based ILs having different alkyl chains that have been found to be effective against *E. coli*, B. subtilis, and S. aureus [17]. Recently, Doria et al. reported *N*-cinnamyl imidazole with four different alkyl chain lengths against pathogenic bacteria (*E. coli*; *P. aeruginosa*; *S. pyogenes*; *S. aureus*; *S. epidermidis*) [3]. S Anvari et al.’s study of pyridinium- and imidazolium-based ILs with different alkyl chains reported that longer-chain anions have more antibacterial potential than smaller-chain anions against the strains of B. subtilis, *E. coli*, *K. pneumoniae*, *S. aureus*, and *S. typhimurium* [18]. In our previous work, octyl-based ionic liquids were more antibacterial than in a recent work based on the butyl chain length that also showed that the short length of the chain is why ionic liquids have less antibacterial activity than in previous work [1]. A study by N Iwai using three different classes: imidazolium, pyrrolidinium, and piperidinium salts, and various alkyl- or silylalkyl-group chain lengths, proved antibacterial properties against *E. coli* [19]. Phosphonium-based ILs with different anions have proven their potential in antibacterial, antioxidant, and anticancer applications in the pharmaceutical industry [20]. Quaternary ammonium-based ILs were found to be significantly effective against *E. coli* [21]. Modified quaternary ammonium-based ILs were found to be very effective against *Staphylococcus aureus* and Monilia albican, and less effective against *Pseudomonas aeruginosa* [22]. F. Walkiewicz et al. synthesized azolate-based tetraammonium ionic liquids; the anions used were 4-nitroimidazolate (4-NO_2_Im), benzotriazolate (Bt), 2-methyl-4-nitroimidazolate (2-Me-4-NO_2_Im), and 1,2,4-triazolate (Tr), and the cations used were hexadecyltrimethylammonium (CTA), domiphen (DOM), DDA, and BA; these were found to be good antibacterial agents [23].

Synthetic procedures of different types of ILs are reported by copious methods [4]. The synthesis of imidazolium-, pyridinium-, and quaternary ammonium-based ionic liquids using 1-methylimidazole, pyridine, and diethyl amine (DEA) by reacting with butyl bromide has attracted interest owing to the advantages of the simple synthetic procedure. We used the most widely accepted method of preparation, which is known as the metathesis method. The resulting ionic liquids had bromide as an anion, while imidazolium, pyridinium, and quaternary ammonium were the cations. To create a library of ILs, these three different classes of lab-synthesized ionic liquids were further converted into three lines of ionic liquids using methanesulfonate, bis(trifluoromethanesulfonyl)imide, dichloroacetate, tetrafluoroborate, and hydrogen sulfate to replace the bromide, as shown in Table 1. These ILs were characterized by FTIR spectroscopy and NMR for structure elucidation and chemical-synthesis confirmation.

The interaction of ILs and cells (prokaryotic or eukaryotic) is the major question to be answered before using ILs for antimicrobial activity. ILs, because of their amphiphilic properties, have the potential to interact with the cellular membrane, producing lipids and leading to the cell bursting. Quaternary ammonium ions, especially, have been reported to change the outer zeta potential of Gram-negative bacteria, and this changes the cellular function. ILs have also been reported to affect the diffusion, fluidity, and permeability of cellular membranes [24]. Therefore, in this study, we used the most common pathogenic bacteria (i.e., five strains of Gram-negative bacteria and two strains of Gram-positive bacteria) with our library of compounds to evaluate the antimicrobial potential of the synthesized ILs. Moreover, we also carried out the modeling of the ionic-liquid antimicrobial activity through FTIR spectral-data results [25]. Herein, we report the synthetic procedures and in vitro antibacterial studies of these synthesized ILs in detail.

The library of the 18 novel ILs based on the butyl chain were prepared to be tested as potential antibacterial drugs. In the first step, cations containing imidazolium, pyridinium, and ammonium cations with butyl chains were prepared. In the next step, bromide (Br^−^), methanesulfonate (CH_3_SO_3_^−^), bis(trifluoromethanesulfonyl)imide (NTf_2_^−^), dichloroacetate (CHCl_2_CO_2_^−^), tetrafluoroborate (BF_4_^−^), and hydrogen sulfate (HSO_4_^−^) anions were substituted.

## 2. Materials and Methods

### 2.1. Chemicals for Synthesis

Chemicals and reagents (viz., 1-methyl imidazole; pyridine; diethylamine (DEA); butyl bromide; methane sulfonic acid; dichloroacetic acid; potassium hydrogen sulfate; lithium bis(trifluoromethanesulfonyl)imide (Li [Tf_2_N]); sodium tetrafluoroborate (Na [BF_4_]); potassium hydrogen sulfate (K[HSO_4_]); sodium hydroxide) were procured from Sigma Aldrich. All solvents, such as methanol (CH_3_OH), chloroform (CHCl_3_), acetonitrile (CH_3_CN), n-Hexane, ethyl acetate, and distilled water, were purchased from Sigma Aldrich. All organic solvents were dried with the help of distillation assembly.

### 2.2. Characterization Techniques

Thin-layer chromatography (TLC) was carried out to check the progress of the reaction. Analytical TLC was performed using Merck-prepared plates (silica gel 60 F-254 on aluminum). The solvent system used to check the completion of the ion-exchange reaction was DCM:MeOH:CH_3_COOH (80:20:5). Spectroscopic methods were used for the chemical-structure elucidation. Infrared spectra were obtained on a Bruker platinum ATR model Alpha spectrophotometer (Germany). A total of 20 mg of sample was scanned from the 4000 to 400 cm^−1^ wave numbers. ^1^HNMR spectra were recorded for all samples at room temperature in deuterated dimethyl sulfoxide (DMSO-d6) on a 400 MHz Bruker AV400 spectrometer (Bruker Corporation, Billerica, MA, USA), with a concentration of 20 mg/mL^−1^. An agar well diffusion assay was carried out to evaluate the antibacterial potential of the prepared samples against pathogenic bacteria. A total of 6 different species/strains of bacteria were tested against 18 ionic-liquid samples. Fresh bacterial culture (24 h) of five Gram-negative strains (*Escherichia coli*; *Klebsiella aerogene/Enterobacter aerogenes*; *Klebsiella pneumoniae*; *Proteus vulgaris*; *Pseudomonas aeruginosa*) and two Gram-positive strains (*Streptococcus pyogenes* and *Streptococcus pneumoniae*) were used in this experiment. Each bacterial culture was freshly prepared in nutrient broth to the logarithmic phase at 37 °C in an orbital shaker from 16 to 18 h. A UV–visible spectrophotometer was used to check the optical density of the culture at 600 nm. When the optical density was between 0.58 and 0.6, working solutions of a 10^6^–10^5^ CFU/mL bacterial concentration from these bacterial cultures were prepared for the antibacterial studies. Samples of equal concentrations and densities were prepared in sterile distilled water. Sterile Petri dishes were inoculated with 100 µL of each equalized bacterial sample to create a uniform lawn on each Petri plate. Using a sterilized borer, 3 mm wells were made at equal distances. Levofloxacin as the positive control and distilled water as the negative control were used in equal volume-by-volume ratios on every plate. About 25 µL of each sample solution was poured into each well and was then incubated for 24 h at 37 °C. Each sample was tested in triplicate.

### 2.3. Synthesis

The three different starting ionic liquids synthesized were 1-butyl-3-methylimidazolium bromide, butylpyridinium bromide, and diethyldibutylammonium bromide, which were synthesized by the metathesis method. The 1-methyl-3-butylimidazolium bromide and butylpyridinium bromide were synthesized according to the literature [26]. Their synthesis methods are briefly described in Section 2.3.

Two anions (sodium methanesulfonate (Na [CH_3_SO_3_]) and sodium dichloroacetate (Na [CHCl_2_CO_2_])) were synthesized in a lab. Their synthesis methods are briefly described in Section 2.4.

### 2.4. Synthesis of Bromide-Based Ionic Liquids

#### 2.4.1. Methyl-3-Butylimidazolium Bromide ([C_4_mim] Br) Synthesis

1-methyl imidazole (5 g, 60 mmol) was reacted with butyl bromide (8.36 g, 61 mmol) in acetonitrile (60 mL) to obtain 1-methyl-3-butylimidazolium bromide (Figure 1), and the reaction mixture was refluxed and stirred for 48 h to obtain the maximum yield of the product. The completion of the reaction was confirmed with the help of TLC, after the reaction completion solvent was evaporated with the help of a rotary evaporator, and the reactants were decanted. Then, the product was dried in vacuum oven for 12 h at 50 °C. The product obtained was a yellow oily liquid with a 92% yield.

#### 2.4.2. Butyl Pyridinium Bromide ([C_4_py] Br) Synthesis

Pyridine (5 g, 63.3 mmol) was reacted with butyl bromide (8.66 g, 63.3 mmol) in acetonitrile (60 mL) to obtain butyl pyridinium bromide. The reaction mixture was refluxed and stirred for 48 h to obtain the maximum yield of the product (Figure 1). After confirmation of the reaction completion with the help of TLC, the reaction was stopped, the solvent was rotary evaporated, and the reactants were decanted to obtain the brown-colored liquid product. The product was further dried in a vacuum oven for 12 h at 50 °C, and the yield was 93%.

#### 2.4.3. Diethyl Dibutyl Ammonium Bromide ([N_2,2,4,4_] Br) Synthesis

Diethyl dibutyl ammonium bromide was synthesized by reacting diethylamine (2.5 g, 34 mmol) with 2 molar equivalents of bromobutane (9.36 g, 68 mmol) in acetonitrile (60 mL). A total of 3 molar equivalents (10.8 g, 102 mmol) of sodium carbonate (Na_2_CO_3_) were taken as the base. The reaction mixture containing reactants, solvent, and base was evacuated with the help of a vacuum pump to remove air, and it was then stirred and refluxed under a nitrogen atmosphere at 90 °C for 24 h to obtain the required product (Figure 1). The completion of the reaction was confirmed with TLC, the solvent was rotary evaporated, the unreacted material was decanted, the base was filtered to obtain pure product, and the pure crystalline product was obtained and dried in a vacuum oven for 12 h at 50 °C; the yield was 93%.

### 2.5. Synthesis of Butyl Imidazolium-Based Ionic Liquids

#### 2.5.1. 1-Butyl-3-Methylimidazolium Methanesulfonate ([C_4_mim] [CH_3_SO_3_H]) Synthesis

1-butyl-3-methylimidazolium methanesulfonate was synthesized with the help of a metathesis reaction of 1-butyl-3-methylimidazolium bromide (2.41 g, 11 mmol) with sodium methanesulfonate (1.3 g, 10 mmol). The reaction mixture was stirred overnight at room temperature in methanol (Figure 2). Sodium bromide was removed from the product with the help of a solvent extraction followed by filtration. Chloroform and ethyl-acetate were used for the solvent extraction of the product from sodium bromide. Several fractions of these solvents containing the product were collected through filtration, and then the solvent was rotary evaporated to obtain a pure product, which was further completely dried in a vacuum oven overnight at 50 °C; the product yield was 78%.

#### 2.5.2. 3-Butyl-1-Methylimidazolium Bis(Trifluoromethanesulfonyl)Imide ([C_4_mim] [Tf_2_N]) Synthesis

A solution of LiTf_2_N (0.313 g, 1 mmol) in 2-neck round-bottom flasks was evacuated with the help of a vacuum pump, an inert atmosphere was created in the flask, and then 3-butyl-1-methylimidazolium bromide (0.3 g, 1 mmol) solution was added to the flask. Two reactants were stirred continuously overnight under an inert atmosphere in methanol (60 mL) (Figure 2). After the product formation, the methanol was evaporated, and the lithium bromide (LiBr) was separated with the help of solvent extraction and filtration, for which chloroform was used, and the chloroform was rotary evaporated. The product formed was dried in a vacuum oven, which resulted in a 78% yield.

#### 2.5.3. 3-Butyl-1-Methylimidazolium Dichloroacetate ([C_4_mim] [CHCl_2_CO_2_]) Synthesis

A solution of [C_4_mim] Br (0.44 g, 2 mmol) was slowly added to a solution of Na [CHCl_2_CO_2_] (0.3 g, 2 mmol) and was stirred overnight in acetonitrile (60 mL) to obtain the product ([C_4_mim] [CHCl_2_CO_2_]) (Figure 2). NaBr was filtered and acetonitrile was evaporated to obtain the product, the remaining NaBr was removed with solvent extraction and filtration, and the filtrate was dried to obtain the pure product.

#### 2.5.4. 1-Methyl-3-Butylimidazolium Tetrafluoroborate ([C_4_mim] [BF_4_]) Synthesis

Solutions of [C_4_mim] Br (0.59 g, 2.6 mmol) and Na [BF_4_] (0.3 g, 2.7 mmol) in acetonitrile (60 mL) were kept at room temperature for 12 h in an inert atmosphere (Figure 2). The NaBr was separated as residue on filter paper and the product was passed as filtrate, and the filtrate was dried through the rotary evaporator to obtain the product, while the remaining NaBr was further removed with the solvent-extraction technique, the solvent was evaporated, and the product was vacuum-dried.

#### 2.5.5. 1-Methyl-3-Butylimidazolium Hydrogen Sulfate ([C_4_mim] [HSO_4_]) Synthesis

[C_4_mim] Br (0.48 g, 2.1 mmol) solution in distilled water was slowly added to a solution of K [HSO_4_] (0.3 g, 2.2 mmol) in distilled water and stirred overnight at room temperature to obtain the product ([C_4_mim] [HSO_4_]) (Figure 2). The solvent was rotary evaporated, and the KBr was separated through solvent extraction followed by filtration. The product was vacuum-dried to obtain the completely dried product.

### 2.6. Synthesis of Butyl Pyridinium-Based Ionic Liquids

#### 2.6.1. Butyl Pyridinium Methanesulfonate ([C_4_py] [CH_3_SO_3_H]) Synthesis

Butyl pyridinium bromide (0.9 g, 4 mmol) and sodium methanesulfonate (0.5 g, 4.2 mmol) were stirred overnight at room temperature in methanol (80 mL) to obtain butyl pyridinium methanesulfonate (Figure 3). After reaction completion, the solvent was rotary evaporated and sodium bromide was solvent-extracted with the help of ethyl-acetate and chloroform. Several fractions of solvent were collected through filtration that contained the product and remaining sodium bromide on filter paper. Fractions of solvent were evaporated, and the product was vacuum-dried.

#### 2.6.2. Butyl Pyridinium Bis(Trifluoromethanesulfonyl)Imide ([C_4_py] [Tf_2_N]) Synthesis

Li [Tf_2_N] (0.3 g, 1 mmol) solution in methanol (30 mL) was formed in 2-neck RB flasks, and it was then evacuated, and an inert atmosphere was created. A solution of [C_4_py] Br (0.39 g, 1.8 mmol) in methanol (30 mL) was added and stirred overnight to form [C_4_py] [Tf_2_N] (Figure 3). The methanol was evaporated, the LiBr with solvent extraction and filtration was separated from the product, the solvent was rotary evaporated to obtain the pure product, and the product was further dried in a vacuum oven at 40 °C overnight.

#### 2.6.3. Butyl Pyridinium Dichloroacetate ([C_4_py] [CHCl_2_CO_2_]) Synthesis

[C_4_py] Br (0.4 g, 1.8 mmol) and Na [CHCl_2_CO_2_] (0.3 g, 2 mmol) solutions were prepared in acetonitrile (60 mL). [C_4_py] [CHCl_2_CO_2_] was synthesized when [C_4_py] Br solution was slowly added to Na [CHCl_2_CO_2_] solution and stirred overnight at room temperature (Figure 3). After product formation, NaBr was separated from the solvent through filtration, and the solvent was rotary evaporated, and further NaBr was separated with solvent extraction followed by filtration. The product was completely dried in a vacuum oven overnight.

#### 2.6.4. Butyl Pyridinium Tetrafluoroborate ([C_4_py] [BF_4_]) Synthesis

[C_4_py] Br (0.59 g, 2.7 mmol) was slowly added to evacuated Na [BF_4_] (0.3 g, 2.75 mmol), and reaction mixture was stirred in acetonitrile (60 mL) overnight in inert atmosphere to obtain [C_4_py] [BF_4_] (Figure 3). NaBr was removed through solvent extraction and filtration, the solvent was evaporated, and the product was completely dried in a vacuum oven overnight.

#### 2.6.5. Butyl Pyridinium Hydrogen Sulfate ([C_4_py] [HSO_4_]) Synthesis

Butyl pyridinium bromide ([C_4_py] Br) (0.4 g, 1.8 mmol) and potassium hydrogen sulfate (K[HSO_4_]) (0.3 g, 2.2 mmol) were added to 30 mL distilled water separately to form a solution, then [C_4_py] Br solution was slowly added to K[HSO_4_] solution and stirred overnight at room temperature to form the product ([C_4_py] [HSO_4_]) and KBr (Figure 3). Water was rotary evaporated, KBr was removed with help of solvent extraction and filtration, and the solvent was rotary evaporated to obtain the product, which was further completely dried in a vacuum oven.

### 2.7. Synthesis of Ammonium-Based Ionic Liquids

#### 2.7.1. Diethyl Dibutyl Ammonium Methanesulfonate ([N_2,2,4,4_] [CH_3_SO_3_H]) Synthesis

Diethyl dibutyl ammonium bromide (1 g, 3.7 mmol) solution in methanol (40 mL) was added to a sodium methanesulfonate (0.44 g, 3.6 mmol) methanol solution (40 mL) and stirred overnight at room temperature to obtain diethyl dibutyl ammonium methanesulfonate (Figure 4). Methanol was evaporated, sodium bromide was solvent-extracted, solvents were rotary evaporated, and product was completely dried in a vacuum oven.

#### 2.7.2. Diethyl Dibutyl Ammonium Bis(Trifluoromethanesulfonyl)Imide ([N_2,2,4,4_] [Tf_2_N]) Synthesis

[N_2,2,4,4_] Br (0.5 g, 1.8 mmol) solution in methanol (30 mL) was added to evacuated solution of Li [Tf_2_N] (0.5 g, 1.8 mmol) in methanol (30 mL) and stirred overnight under inert atmosphere at room temperature (Figure 4). The product ([N_2,2,4,4_] [Tf_2_N]) and LiBr were dried in a rotary evaporator to evaporate the methanol, then the LiBr was separated with solvent extraction and filtration, and the filtrate containing the product was dried from the solvent to obtain the pure and dried product.

#### 2.7.3. Diethyl Dibutyl Ammonium Dichloroacetate ([N_2,2,4,4_] [CHCl_2_CO_2_]) Synthesis

Diethyl dibutyl ammonium dichloroacetate ([N_2,2,4,4_] [CHCl_2_CO_2_]) was obtained when Na [CHCl_2_CO_2_] (0.28g, 1.8 mmol) solution in acetonitrile (40 mL) was added to evacuated solution of [N_2,2,4,4_] Br (0.5 g, 1.8 mmol) in acetonitrile (40 mL), and reaction mixture was stirred under nitrogen atmosphere at room temperature overnight (Figure 4). The NaBr was filtered, the solvent was dried, the pure product was obtained using the solvent-extraction technique, and the product was then dried completely.

#### 2.7.4. Diethyldibutylammonium Tetrafluoroborate ([N_2,2,4,4_] [BF_4_]) Synthesis

Diethyldibutylammonium tetrafluoroborate ([N_2,2,4,4_] [BF_4_]) was synthesized by stirring solutions of diethyldibutylammonium bromide ([N_2,2,4,4_] Br) (1 g, 3.7 mmol) and sodium tetrafluoroborate (Na[BF_4_]) (0.4g, 3.6 mmol) in acetonitrile (80 mL) under an inert and evacuated atmosphere overnight at room temperature (Figure 4). The sodium bromide was filtered, the solvent was evaporated, and the product was completely dried in a vacuum oven overnight.

#### 2.7.5. Diethyl Dibutyl Ammonium Hydrogen Sulfate ([N_2,2,4,4_] HSO_4_]) Synthesis

[N_2,2,4,4_] Br (1 g, 3.7 mmol) solution in distilled water (30 mL) was slowly added to a solution of K[HSO_4_] (0.5, 3.6 mmol) in distilled water (30 mL) and stirred overnight at room temperature to obtain the product ([N_2,2,4,4_] [HSO_4_]) (Figure 4). Solvent was rotary evaporated, and KBr was separated through solvent extraction followed by filtration. The product was vacuum-dried to obtain a completely dried product.

### 2.8. Spectroscopic Data of Syntheses

#### 2.8.1. 1-Methyl-3-Butylimidazolium Bromide

**Yield:** 92%. **FTIR:** 3120 cm^−1^; 2958 cm^−1^; 2854 cm^−1^; 1650 cm^−1^; 1567 cm^−1^; 1463 cm^−1^; 1165 cm^−1^; 752 cm^−1^. **^1^H NMR:**
^1^H NMR ppm (CDCl_3_, ppm): 0.88–0.92 (m, 3H; CH_3_); 1.28–1.32 (m, 2H; CH_2_); 1.70–1.90 (m, 2H; CH_2_); 3.9–4.1 (m, 3H; NCH_3_); 4.18–4.32 (m, 2H; NCH_2_); 7.40–7.49 (s, 1H; CH=CH); 7.50–7.60 (s, 1H; CH=CH); 9.90–10.10 (s, 1H; N-CH-N). ^13^C NMR ppm: 13.8 (CH_3_); 20.7 (NCH_2_); 32.3 (NCH_2_); 37.1 (CH_3_); 54.2 (NCH2); 122 (CH); 123.0 (CH); 137.0 (CH).

#### 2.8.2. Butyl Pyridinium Bromide

**Yield:** 93%. **FTIR:** 3120 cm^−1^; 2960 cm^−1^; 2855 cm^−1^; 1632 cm^−1^; 1569 cm^−1^; 1486 cm^−1^; 1170 cm^−1^; 770 cm^−1^. **^1^H NMR spectra to confirm synthesis of [C_4_py]Br:** ^1^H NMR ppm (CDCl_3_, ppm): 0.85–0.91 (m, CH_3_); 1.28–1.32 (m, 2H;CH_2_); 1.90–1.99 (m, 2H;CH_2_); 4.88–4.99 (m, 2H;NCH_2_); 8.10–8.20 (m, 1H; CH-CH); 8.55–8.65 (m, 1H; CHCHCH); 9.45–9.55 (m, 1H; NCH). ^13^C NMR ppm: 13.8 (CH_3_); 21.1 (CH_2_); 32.6 (CH_2_); 71.6 (CH_2_); 128.4 (CH); 129.0 (CH); 142.5 (CH); 142.5 (CH); 148.4 (CH).

#### 2.8.3. Diethyl Dibutyl Ammonium Bromide

**Yield:** 93%. **FTIR**: 2960 cm^−1^; 2854 cm^−1^; 1463 cm^−1^; 1395 cm^−1^; 1160 cm^−1^; 743 cm^−1^. ^1^HNMR (400 MHz, DMSO-d6) δ ppm: 3.21–3.35 (m, 8H); 1.71 (quin, 4H); 1.22–1.33 (m, 10H); 0.89 (t, 6H). ^13^C NMR (100 MHz, DMSO-d6): 8.34 (-C-CH_3_); 13.86 (CH_3_); 19.67 (CH_2_); 23.58 (CH_2_); 55.63 (CH_2_); 57.25 (CH_2_).

#### 2.8.4. 1-Butyl-3-Methylimidazolium Methanesulfonate

**Yield:** 78%. **FTIR**: 3120 cm^−1^; 2958 cm^−1^; 2854 cm^−1^; 1650 cm^−1^; 1567 cm^−1^; 1463 cm^−1^; 1200 cm^−1^; 1165 cm^−1^; 752 cm^−1^. ^1^HNMR (400 MHz, DMSO-d6) δ ppm: 0.90 (t, -C-CH_3_); 1.30–1.39 (m, CH_2_); 2.02 (quin, CH_2_); 2.84 (s, -S-CH_3_); 3.72 (s, N-CH_3_); 5.03 (t, 2H); 7.91 (d, 1H); 7.77 (d, 1H); 8.93 (s, -N-CH-N-). ^13^C NMR (100 MHz, DMSO-d6): 13.85 (-C-CH_3_); 37.15 (-N-CH_3_); 44.5 (-S-CH_3_); 54.27 (N-CH_2_); 122.84 (aromatic); 123.21 (aromatic); 137.02 (aromatic).

#### 2.8.5. 3-Butyl-1-Methylimidazolium Bis(Trifluoromethanesulfonyl)Imide

**Yield:** 78%. **FTIR**: 2929 cm^−1^; 2854 cm^−1^; 1569 cm^−1^; 1463 cm^−1^; 1346 cm^−1^; 1180 cm^−1^; 1165 cm^−1^; 740 cm^−1^. ^1^HNMR (400 MHz, DMSO-d6) δ ppm: 0.91 (t, -C-CH_3_); 1.31–1.41 (m, CH_2_); 2.03 (quin, CH_2_); 3.72 (s, N-CH_3_); 5.05 (t, 2H); 7.77 (d, 1H); 7.92 (d, 1H); 8.94 (s, -N-CH-N-). ^13^C NMR (100 MHz, DMSO-d6): 13.87 (-C-CH_3_); 37.19 (-N-CH_3_); 54.29 (N-CH_2_); 122.82 (aromatic); 123.20 (aromatic); 137.02 (aromatic); 149.23 (F-C-S-).

#### 2.8.6. 3-Butyl-1-Methylimidazolium Dichloroacetate

**Yield:** 85%. **FTIR:** 2928 cm^−1^; 2854 cm^−1^; 1633 cm^−1^; 1569 cm^−1^; 1463 cm^−1^; 1375 cm^−1^; 1165 cm^−1^; 740 cm^−1^. ^1^HNMR (400 MHz, DMSO-d6) δ ppm: 0.89 (t, -C-CH_3_); 1.30–1.39 (m, CH_2_); 2.02 (quin, CH_2_); 3.72 (s, N-CH_3_); 5.02 (t, 2H); 6.32 (s, -CH-Cl); 7.75 (d, 1H); 7.92 (d, 1H); 8.92 (s, -N-CH-N-). ^13^C NMR (100 MHz, DMSO-d6): 13.87 (-C-CH_3_); 20.73, 32.32, 37.19 (-N-CH_3_); 54.26 (N-CH_2_); 76.86 (-C-Cl); 122.81 (aromatic); 123.22 (aromatic); 137.04 (aromatic); 160.39 (CO).

#### 2.8.7. 1-Methyl-3-Butylimidazolium Tetrafluoroborate

**Yield:** 70%. **FTIR**: 3120 cm^−1^; 2924 cm^−1^; 2854 cm^−1^; 1573 cm^−1^; 1463 cm^−1^; 1169 cm^−1^; 752 cm^−1^. ^1^HNMR (400 MHz, DMSO-d6) δ ppm: 0.89 (t, -C-CH_3_); 1.29–1.38 (m, CH_2_); 2.03 (quin, CH_2_); 3.72 (s, -CH_3_); 5.02 (t, 2H); 7.73 (d, 1H); 7.92 (d, 1H); 8.94 (s, -N-CH-N-). ^13^C NMR (100 MHz, DMSO-d6): 13.86 (-C-CH_3_); 32.33, 20.72, 37.17 (-N-CH_3_); 54.23 (N-CH_2_); 122.84 (aromatic); 123.21 (aromatic); 137.02 (aromatic).

#### 2.8.8. 1-Methyl-3-Butylimidazolium Hydrogen Sulfate

**Yield:** 75%. **FTIR**: 3120 cm^−1^; 2958 cm^−1^; 2854 cm^−1^; 1567 cm^−1^; 1463 cm^−1^; 1165 cm^−1^; 1047 cm^−1^; 752 cm^−1^. ^1^HNMR (400 MHz, DMSO-d6) δ ppm: 0.89 (t, -C-CH_3_); 1.30–1.39 (m, CH_2_); 2.01 (quin, CH_2_); 3.72 (s, -CH_3_); 5.01 (t, 2H); 7.72 (d, 1H); 7.94 (d, 1H); 8.5 (bs, -OH); 8.93 (s, -N-CH-N-). ^13^C NMR (100 MHz, DMSO-d6): 13.89 (-C-CH_3_); 20.74, 32.32, 37.17 (-N-CH_3_); 54.22 (N-CH_2_); 122.81 (aromatic); 123.24 (aromatic); 137.02 (aromatic).

#### 2.8.9. Butyl Pyridinium Methanesulfonate

**Yield:** 91%. **FTIR**: 3120 cm^−1^; 2960 cm^−1^; 2855 cm^−1^; 1632 cm^−1^; 1569 cm^−1^; 1486 cm^−1^; 1170 cm^−1^; 1042 cm^−1^; 770 cm^−1^. ^1^HNMR (400 MHz, DMSO-d6) δ ppm: 0.89 (t, 3H); 1.28–1.33 (m, 2H); 2.01 (quin, 2H); 2.84 (s, -S-CH_3_); 5.03 (t, 2H); 8.25 (dd, 2H*meta*); 8.74 (d, 1H*para*); 8.90 (d, 2H*ortho*). ^13^C NMR (100 MHz, DMSO-d6): 13.98 (-C-CH_3_); 21.25(CH_2_); 71.47(CH_2_); 44.8 (-S-CH_3_); 128.45 (aromatic); 146.17 (aromatic); 147.01 (aromatic).

#### 2.8.10. Butyl Pyridinium Bis(Trifluoromethanesulfonyl)Imide

**Yield:** 75%. **FTIR**: 1636 cm^−1^; 1569 cm^−1^; 1172 cm^−1^; 1346 cm^−1^; 1051 cm^−1^; 740 cm^−1^. ^1^HNMR (400 MHz, DMSO-d6) δ ppm: 0.89 (t, 3H); 1.26–1.33 (m, 2H); 2.04 (quin, 2H); 5.02 (t, 2H); 8.24 (dd, 2H*meta*); 8.76 (d, 1H*para*); 8.90 (d, 2H*ortho*). ^13^C NMR (100 MHz, DMSO-d6): 13.98 (-C-CH_3_); 21.25 (CH_2_); 71.47 (CH_2_); 128.49 (aromatic); 146.83 (aromatic); 147.45 (aromatic); 148.83 (F-C-S-).

#### 2.8.11. Butyl Pyridinium Dichloroacetate

**Yield:** 90%. **FTIR**: 2962 cm^−1^; 2855 cm^−1^; 1633 cm^−1^; 1569 cm^−1^; 1486 cm^−1^; 1370 cm^−1^; 1172 cm^−1^; 769 cm^−1^. ^1^HNMR (400 MHz, DMSO-d6) δ ppm: 0.89 (t, 3H); 1.27–1.33 (m, 2H); 2.04 (quin, 2H); 5.04 (t, 2H); 6.35 (s, -CH-Cl); 8.24 (dd, 2H*meta*); 8.74 (d, 1H*para*); 8.89 (d, 2H*ortho*). ^13^C NMR (100 MHz, DMSO-d6): 13.98 (-C-CH_3_); 21.25 (CH_2_); 71.47 (CH_2_); 76.83 (-C-Cl); 128.45 (aromatic); 146.15 (aromatic); 146.99 (aromatic); 162.14 (CO).

#### 2.8.12. Butyl Pyridinium Tetrafluoroborate

**Yield:** 88%. **FTIR**: 2926 cm^−1^; 2855 cm^−1^; 1635 cm^−1^; 1569 cm^−1^; 1489 cm^−1^; 1172 cm^−1^; 1049 cm^−1^; 771 cm^−1^. ^1^HNMR (400 MHz, DMSO-d6) δ ppm: 0.89 (t, 3H); 1.27–1.31 (m, 2H); 2.01 (quin, 2H); 5.01 (t, 2H); 8.21 (dd, 2H*meta*); 8.72 (d, 1H*para*); 8.85 (d, 2H*ortho*). ^13^C NMR (100 MHz, DMSO-d6): 13.98 (-C-CH_3_); 21.25 (CH_2_); 71.47 (CH_2_); 128.43 (aromatic); 145.54 (aromatic); 146.94 (aromatic).

#### 2.8.13. Butyl Pyridinium Hydrogen Sulfate

**Yield: 93%. FTIR**: 3120 cm^−1^; 2923 cm^−1^; 2854 cm^−1^; 1632 cm^−1^; 1569 cm^−1^; 1486 cm^−1^; 1169 cm^−1^; 1046 cm^−1^; 774 cm^−1^. ^1^HNMR (400 MHz, DMSO-d6) δ ppm: 0.89 (t, 3H); 1.27–1.33 (m, 2H); 2.01 (quin, 2H); 5.02 (t, 2H); 8.23 (dd, 2H*meta*); 8.45 (bs, -OH); 8.71 (d, 1H*para*); 8.85 (d, 2H*ortho*). ^13^C NMR (100 MHz, DMSO-d6): 13.98 (-C-CH_3_); 21.25 (CH_2_); 71.47 (CH_2_); 128.43 (aromatic); 146.14 (aromatic); 146.96 (aromatic).

#### 2.8.14. Diethyl Dibutyl Ammonium Methane Sulfonate

**Yield: 76%. FTIR**: 2960 cm^−1^; 2854 cm^−1^; 1459 cm^−1^; 1395 cm^−1^; 1202 cm^−1^; 1038 cm^−1^; 743 cm^−1^. ^1^HNMR (400 MHz, DMSO-d6) δ ppm: 0.89 (t, 6H); 1.21–1.32 (m, 10H); 1.72 (quin, 4H); 2.84 (s, -S-CH_3_); 3.21–3.36 (m, 8H). ^13^C NMR (100 MHz, DMSO-d6): 8.34 (-C-CH_3_); 13.86 (CH_3_); 19.64 (CH_2_); 23.58 (CH_2_); 44.59 (-S-CH_3_); 55.63 (CH_2_); 57.25 (CH_2_).

#### 2.8.15. Diethyl Dibutyl Ammonium Bis(Trifluoromethanesulfonyl)Imide

**Yield: 78%. FTIR**: 2960 cm^−1^; 2854 cm^−1^; 1459 cm^−1^; 1395 cm^−1^; 1348 cm^−1^; 1052 cm^−1^; 739 cm^−1^. ^1^HNMR (400 MHz, DMSO-d6) δ ppm: 0.89 (t, 6H); 1.23–1.35 (m, 10H); 1.73 (quin, 4H); 3.23–3.38 (m, 8H). ^13^C NMR (100 MHz, DMSO-d6): 8.34 (-C-CH_3_); 13.86 (CH_3_); 19.67 (CH_2_); 23.58 (CH_2_); 55.63 (CH_2_); 57.25 (CH_2_); 149.35 (F-C-S).

#### 2.8.16. Diethyl Dibutyl Ammonium Dichloroacetate

**Yield:** 83%. **FTIR**: 2961 cm^−1^; 2854 cm^−1^; 1459 cm^−1^; 1370 cm^−1^; 1646 cm^−1^; 1160 cm^−1^; 710 cm^−1^. ^1^HNMR (400 MHz, DMSO-d6) δ ppm: 0.89 (t, 6H); 1.23–1.35 (m, 10H); 1.73 (quin, 4H); 3.21–3.35 (m, 8H); 6.35 (s, -CH-Cl). ^13^C NMR (100 MHz, DMSO-d6): 8.34 (-C-CH_3_); 13.86 (CH_3_); 19.67 (CH_2_); 23.58 (CH_2_); 55.63 (CH_2_); 57.25 (CH_2_); 76.57 (-CH-Cl); 162.91 (CO).

#### 2.8.17. Diethyl Dibutyl Ammonium Tetrafluoroborate

**Yield:** 75%. **FTIR**: 2960 cm^−1^; 2854 cm^−1^; 1459 cm^−1^; 1393 cm^−1^; 1040 cm^−1^; 751 cm^−1^. ^1^HNMR (400 MHz, DMSO-d6) δ ppm: 0.89 (t, 6H); 1.24–1.33 (m, 10H); 1.72 (quin, 4H); 3.22–3.33 (m, 8H). ^13^C NMR (100 MHz, DMSO-d6): 8.34 (-C-CH_3_); 13.86 (CH_3_); 19.67 (CH_2_); 23.58 (CH_2_); 55.63 (CH_2_); 57.25 (CH_2_).

#### 2.8.18. Diethyl Dibutyl Ammonium Hydrogen Sulfate

**Yield:** 73%. **FTIR**: 2960 cm^−1^; 2854 cm^−1^; 1459 cm^−1^; 1393 cm^−1^; 1160 cm^−1^; 1027 cm^−1^; 739 cm^−1^. ^1^HNMR (400 MHz, DMSO-d6) δ ppm: 0.89 (t, 6H); 1.24–1.33 (m, 10H); 1.72 (quin, 4H); 3.22–3.33 (m, 8H). ^13^C NMR (100 MHz, DMSO-d6): 8.34 (-C-CH_3_); 13.86 (CH_3_); 19.67 (CH_2_); 23.58 (CH_2_); 55.63 (CH_2_); 57.25 (CH_2_).

### 2.9. Computational Details

#### 2.9.1. Density Functional Theory

All eighteen ionic liquids were modeled and optimized using Gaussian 09 [27]. The geometry optimizations were performed using the functional B3LYP/6-31g level of the density-functional-theory methods [28]. For validation purposes, the frequencies [29] were calculated for the optimized geometries at the same energy level.

#### 2.9.2. Molecular-Docking Studies

For molecular docking, optimized ionic structures were used. The imidazolium-, pyridinium-, and quaternary ammonium-based ionic liquids were docked against six strains of Gram-positive and Gram-negative bacteria (*E. coli* (PDB ID: 5A924) [30]; *K. pneumonia* (PDB ID: 6MGX6); *E. aerogenes* (PDB ID: 5KID5) [31]; *P. aeruginosa* (PDB ID: 4GZB8) [32]; *P. vulgaris* (PDB ID: 1HZO7) [33]; *S. pneumoniae* (PDB ID: 1RPS9) [34]) using PyRx software. As per the requirements, before docking, energy minimization was performed for all proteins where water molecules and protonations were removed. Docking was performed by maintaining default settings for all parameters. Each docked pair of protein and ligand generated nine different poses, out of which the pose having the highest value of binding affinity was chosen.

## 3. Results and Discussion

Three different cations (1-butyl-3-methylimidazolium bromide, butylpyridinium bromide, and diethyldibutylammonium bromide) were synthesized by the metathesis method [35], which is also known as the green method of synthesis. The 1-methyl-3-butylimidazolium bromide and butylpyridinium bromide were synthesized according to the literature [11]. Three different series of the ionic liquids were prepared: imidazolium-based, butylpyridinium-based, and ammonium-based ionic liquids. All the samples were characterized by spectroscopic data, and their in vivo antibacterial potentials were carried out.

### 3.1. FTIR Analysis

The FTIR for the 1-methyl-3-butylimidazolium bromide cation indicated the quaternarization of methyl imidazole with butyl bromide at 3120 cm^−1^ (aromatic CH stretch), 2958 cm^−1^ and 2854 cm^−1^ (aliphatic CH stretch) CH_2_ and CH_3_, respectively, 1650 cm^−1^ (C=N stretch), 1567 cm^−1^ (C=C stretch), 1463 cm^−1^ (CH_2_ bend), 1165 cm^−1 (^C-N stretch), and 752 cm^−1^ (long-chain CH_2_ of octyl group) [36]. The formation of 1-methyl-3-butylimidazolium methane sulfonate was indicated by the FTIR band appearance of S=O at 1200 cm^−1^. The 1-methyl-3-butylimidazolium bis(trifluoromethanesulfonyl)imide ([C_8_mim] [Tf_2_N]) formation was indicated by FTIR due to the appearance of the band at 1346 cm^−1^, indicating the asymmetric stretching vibration of S=O, and at 1180 cm^−1^, the symmetric stretching vibration of S=O [37]. The band at 1633 cm^−1^ indicated the stretching vibration of C=O, and the band at 1375 cm^−1^ showed the stretching vibration for C-O; these bands indicated the formation of 3-butyl-1-methylimidazolium dichloroacetate ([C_4_mim] [CHCl_2_CO_2_]). In the case of 1-methyl-3-butylimidazolium tetrafluoroborate ([C_4_mim] [BF_4_] I), successful synthesis is indicated by the band appearance at 1050 cm^−1^ of 1-methyl-3-butylimidazolium hydrogen, and the sulfate synthesis was confirmed by the IR band appearance of S=O at 1047 cm^−1^.

The FTIR for the butylpyridinium bromide cation indicates the quaternarization of pyridine with butyl bromide at 3120 cm^−1^ (aromatic CH stretch), 2960 cm^−1^ and 2855 cm^−1^ (aliphatic CH stretch) CH_2_ and CH_3_, respectively, 1632 cm^−1^ (C=N stretch), 1569 cm^−1^ (C=C stretch), 1486 cm^−1^ (CH_2_ bend), 1170 cm^−1^ (C-N stretch), and 770 cm^−1^ (long-chain CH_2_ of butyl group) [38,39].

The IR spectrum of the butyl pyridinium methane sulfonate S=O stretching vibration at 1042 cm^−1^ shows the exchange of the methane sulfonate anion. The FTIR for the butyl pyridinium bis(trifluoromethanesulfonyl)imide ([C_4_mim] [Tf_2_N]) bands at 1346 cm^−1^ and 1051 cm^−1^ are the stretching vibrations for the S=O, indicating anion exchange and confirming a successful synthesis reaction. In the case of butylpyridinium bromide, the band appearing at 1633 cm^−1^ shows the C=O stretching vibration, and the band appearing at 1370 cm^−1^ is for the stretching vibration of C-O, indicating product formation. The strong band appearance at 1049 cm^−1^ confirms the presence of the BF_4_ anion, thus indicating the formation of butyl pyridinium tetrafluoroborate ([C_4_py] [BF_4_]) [40]. At 1046 cm^−1^, the band appearance shows the stretching vibration of the S=O of the HSO_4_ anion, which indicates the formation of butyl pyridinium hydrogen sulfate.

The FTIR for dibutyldiethylammonium bromide ([N_2, 2, 4, 4_] Br) has inherent bands at 2960 cm^−1^ and 2854 cm^−1^ (CH stretch), 1463 cm^−1^ 1395 cm^−1^ (CH_2_ and CH_3_ bends), 1160 cm^−1^ (C-N stretch), and 743 cm^−1^ (CH_2_ in butyl and ethyl groups) [41,42]. The FTIR for the dibutyldiethylammonium methanesulfonate ([N_2, 2, 4, 4_] [MeSO_3_]) at the 1038 cm^−1^ band appearance shows the stretching vibration of the S=O of methanesulfonate. The FTIR for the dibutyldiethylammonium bis(trifluoromethanesulfonyl)imide ([N_2, 2, 4, 4_] [Tf_2_N]) synthesis was confirmed by the band appearance at the 1348 cm^−1^ and 1052 cm^−1^ stretching vibrations of S=O. In the case of dibutyldiethylammonium dichloroacetate ([N_2, 2, 4, 4_] [CHCl_2_CO_2_]), the IR-spectrum strong band at 1646 cm^−1^ is due to the C=O stretching vibrations of the dichloroacetate anion. The strong band appearance at 1049 cm^−1^ confirms the presence of the BF_4_ anion, indicating the formation of 1-methyl-3-octylimidazolium tetrafluoroborate ([C_8_mim] [BF_4_]). The IR spectrum of Di-butyl di-ethyl ammonium hydrogen sulfate ([N_2, 2, 4, 4_] [HSO_4_]) has a 1027 cm^−1^ band appearance due to the S=O stretching vibrations of the hydrogen sulfate anion [43].

These IR spectra confirmed the successful synthesis of the new imidazolium-, butyl pyridinium bromide-, and ammonium-based ionic liquids. Furthermore, these ILs were analyzed by NMR spectroscopy.

### 3.2. HNMR Analysis of Ionic Liquids

The ^1^HNMRs of the ILs were recorded to understand the structure of the synthesis products by evaluating the environments of the protons in the ILs. The HNMRs of the imidazolium-based ionic liquids showed characteristic chemical-shift values of the 1-methyl-3-butylimidazolium bromide [44] cation, as mentioned in Section 2.7. Further changing the bromide ions in the imidazolium-based ILs had almost similar HNMR spectra, having a small shifting of the chemical-shift values for 3-butyl-1-methylimidazolium bis(trifluoromethanesulfonyl)imide and 3-butyl-1-methylimidazolium tetrafluoroborate, while some new chemical-shift values were also observed, which were as follows: 1-butyl-3-methylimidazolium methanesulfonate CH_3_ hydrogens appeared as singlets at 2.84 ppm; 3-butyl-1-methylimidazolium dichloroacetate had 6.32 ppm singlets for -CH-Cl; for the 1-methyl-3-butylimidazolium hydrogen sulfate, a small singlet appeared at 8.5 ppm [45].

Butyl pyridinium bromide has the characteristic ^1^HNMR [46], as presented in Section 2.7. In the case of the butylpyridinium bromide methanesulfonate, singlets of the methyl group of methanesulfonate appeared at 2.84 ppm, while CH-Cl of the butylpyridinium dichloroacetate was observed at 6.35 ppm, and the hydrogen of the OH group of the sulfonic part in the butyl pyridinium hydrogen sulfate had much less intensity (around 8.45 ppm), while there was only a small change in the chemical-shift values of the butylpyridinium bis(trifluoromethanesulfonyl)imide. Butylpyridinium tetrafluoroborate was found, which may be due to the close vicinity of the electronegative atoms.

The newly synthesized diethyldibutylammonium bromide, as shown in the HNMR peaks, is depicted in Section 2.7. Further modification into the other quaternary ammonium-based ILs had the following additions in the quaternary ammonium spectra. The diethyl dibutyl ammonium methanesulfonate had a singlet at 2.84 ppm that corresponded to the CH_3_ present in the methanesulfonate, and the diethyldibutylammonium dichloroacetate had a very strong downshift singlet (CH-Cl) appearing at 6.35 ppm because of the electronegative atom (Cl) attached to it. A very small singlet of OH of hydrogen sulfate in the diethyldibutylammonium hydrogen sulfate appeared downshifted at 8.53 ppm. In the cases of the diethyldibutylammonium bis(trifluoromethanesulfonyl)imide and diethyldibutylammonium tetrafluoroborate, no new chemical-shift values were observed, but the chemical-shift values appeared downshifted because of the electronegativity of the fluorine atom.

The NMR spectroscopic data corroborates the FTIR data and strongly confirms the synthesis of the new ILs, and especially the new series of quaternary ammonium-based ionic liquids that were successfully synthesized. Wang J et al. have also reported similar ^1^HNMR chemical-shift values for ILs [45].

### 3.3. Antibacterial Activities of Ionic Liquids

Encouraged by our work [11,25], we evaluated the antibacterial potential of the newly synthesized ILs. We evaluated the in vitro antibacterial activity by the agar well diffusion method, and the zones of inhibition were measured in 16 hr time, as shown in Figure 1. All the experiments were performed in triplicate, and the results are shown as average values in Figure 1. Levofloxacin as the control was found to be effective against all the bacterial strains of the five Gram-negative bacterial stains (Escherichia *coli*; *Klebsiella aerogene/Enterobacter aerogenes*; *Klebsiella pneumoniae*; *Proteus vulgaris*; *Pseudomonas aeruginosa*) and two Gram-positive strains (*Streptococcus pyogenes* and *Streptococcus pneumoniae*). All the imidazolium-based ionic liquids showed the highest antibacterial activity against all the Gram-negative bacteria and Gram-positive bacteria (Figure 1a) [47]. All the butylpyridinium-based ionic liquids showed antibacterial activity against only one Gram-negative bacterium (*Enterobacter aerogenes*) (Figure 1b) [48], while no activity was found by any of the other butylpyridinium-based ionic liquids, which might be because of the small alkyl chain used in the case of our ILs, while longer chains have been found to be effective against bacteria elsewhere [49]. The ammonium-based ionic liquids (Figure 1c) diethyldibutylammonium tetrafluoroborate and diethyldibutylammonium hydrogen sulfate showed effectiveness against two Gram-negative bacteria (*Proteus vulgaris* and *Klebsiella pneumoniae*), while no activity was found against the other strains. These results are quite convincing and promising because ILs with butyl chains are showing positive and better antibacterial effects against pathogenic Gram-negative bacteria to ILs [18,50,51]. Our results are also in accordance with the PLS data calculated by using FTIR [25]. Borkowski, A et al. has reported that the *E coli* bacterial cultures become adapted and start growing in the presence of tetradecyltrimethylammonium theophyllinate (quaternary ammonium ILs) because quaternary ammonium changes the lipid-membrane structure and protein patterns responsible for bacterial cell death. Bacterial adaptation was responsible for showing antagonistic action with other bactericides [24]. Only imidazolium-based ILs were effective against the Gram-positive bacteria *streptococcus pyogenes* and *streptococcus pneumoniae*, while pyridinium and quaternary ammonium were found to be ineffective against Gram-positive bacteria. This can be explained by the fact that the antibacterial potential is dependent on the unique combination of the cations and anions in the ILs. It was also found that the anions here, such as BF_4_^−^ and HSO_4_^−^, also change the antibacterial potential of quaternary ammonium ILs against *Proteus vulgaris* and *Klebsiella pneumoniae.*

### 3.4. In Silico Antibacterial Activity

The in silico antibacterial activity could be analyzed by calculating the protein–ligand binding energy. The minimum energy value shows a strong protein–ligand binding energy, which means strong inhibition. We performed in silico analyses on all 18 ionic liquids with the Beta Lactamase protein of six bacteria: *E. coli* (PDB ID: 5A92); *K. pneumonia* (PDB ID: 6MGX); *E. aerogenes* (PDB ID: 5KID); *P. aeruginosa* (PDB ID: 4GZB); *P. vulgaris* (PDB ID: 1HZO); *S. pneumoniae* (PDB ID: 1RPS). The results are presented in Table 2.

According to the computational analysis, the imidazolium-based ionic liquids showed binding-affinity values ranging from −0.8 to −7.9 kcal/mol for all the Gram-positive and Gram-negative bacterial strains analyzed. Comparing all six imidazolium-based ionic liquids showed that the bistrifluoromethane and sulfonylimide (Tf_2_N^−^) anions containing ionic liquid showed the maximum inhibitions for all six bacterial strains, with the maximum for *E. aerogenes* (−7.9 kcal/mol), and the minimum for *P. aeruginosa* and *S. pneumonia* (both −5.9 kcal/mol) (Table 2, Figure 2).

Upon comparing the in silico-analysis results for the pyridinium-based ionic liquids, it was observed that the bistriflimide (Tf_2_N^−^) anion containing ionic liquid showed the maximum inhibition for all six bacterial strains (from −7.1 kcal/mol for *K. pneumonia* to −8.5 kcal/mol for *S. pneumonia*) (Table 2). It was also observed that the binding energies for the (Tf_2_N^−^) anion containing pyridinium-based ionic liquid for *E. aerogenes* (−8.3 kcal/mol) and *S. pneumonia* (−8.5 kcal/mol) were similar, which shows that the (Tf_2_N^−^) anion containing pyridinium-based ionic liquid had a similar effect on both strains.

Upon analyzing the molecular-docking results for the quaternary ammonium-based ionic liquids, it was observed that the dicholoroacetate anion containing ionic liquid showed the maximum inhibition for all six bacterial strains, with the maximum against *E. aerogenes* and *S. pneumonia* (−5.5 and −5.4 kcal/mol, respectively). For the other four bacterial strains, it showed the same binding energy (−4.6 kcal/mol) (Table 2).

From Figure 2 (S1-03-ILPs), we interpret that the imidazole sulfonylimide (Tf_2_N^−^) docked with the Ser237, Ser 70, and Ser130 amino acids in the case of *P. vulgaris*, with Ser 315, Asn153, Ser65, and Tyr151 in the case of *E. aerogenes*, and with Asn220 and His 250 in the case of *K. Paneumoniae* were docked with the oxygen of (Tf_2_N^−^) for the pyridinium-based ILs against the bacteria *E. aerogenes*. Sulfonylimide (Tf_2_N^−^) was found docked with Ser 237, as shown in Figure 2 (S2-03-ILP2).

The stabilities of the docked ligands with these bacteria were also observed through the binding interactions between S1-Tf_2_N and the amino acids present at the active sites. The docking of S1-Tf_2_N with *E. aerogenes*, *P. vulgaris*, and *K. pneumoniae* showed the stable hydrophilic interaction of the ligand with the neighboring amino acid residues; Tf_2_N forms the most stable hydrogen-bond interactions (binding affinity: −7.9 kcal/mol) with Ser65, Tyr151, Asn153, and Ser315 when docked with *E. aerogenes*; with *P. vulgaris*, the Tf_2_N showed interactions with Ser70, Ser130, and Ser237, whereas S1-Tf_2_N showed hydrogen bonding with Asn220 and His 250 when docked with *K. pneumoniae*. There were no ligand interactions observed with *E.*
*coli* and *P. aeruginosa*. All these interactions indicated the partial positive nature of the binding sites, as in all cases, and the anionic part of the ionic liquids showed interaction with the surrounding amino acid residues.

## 4. Conclusions

Three different series of ILs using three cations (imidazolium, pyridinium, and quaternary ammonium) and six anions (bromide; methane sulfonate; bis(trifluoromethanesulfonyl)imide; dichloroacetate; tetrafluoroborate; hydrogen sulfate) are reported for the first time. Spectroscopic data were assessed for the structure confirmation of the ILs. The antimicrobial potentials of the ILs were evaluated against the most common pathogenic bacteria that cause infections in wounds. Within 24 h of contact with the bacteria, we found that the imidazolium cation was effective against most of the bacterial strains (with all anions), while the pyridinium-based ILs with all anions were effective against *Enterobacter aerogenes*, the quaternary ammonium with BF_4_^−^ was effective against *Proteus vulgaris*, and HSO_4_^−^ added antibacterial potential to the quaternary ammonium against *Proteus vulgaris* and *Klebsiella pneumoniae*. Our studies are proving that ILs are a good choice in the pharmaceutical industry for antibiotics and antiseptics. According to the molecular-docking results, imidazolium- and pyridinium-based ionic liquids containing the bistriflimide (Tf_2_N^−^) anion, whereas the quaternary ammonium-based ionic liquid containing the dichloroacetate (CHCl_2_CO_2_^−^) anion showed the maximum inhibition against all six bacterial stains. Upon comparing the overall results, it was observed that the bistriflimide (Tf_2_N^−^) containing pyridinium- and imidazolium-based ionic liquids could be a good choice to be used as antibiotics or antiseptics; however, if longer chains (octyl based) are taken, then this would enhance the antibacterial activity.

## Data Availability

Not applicable.

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
