# Peer review of "Synthesis, Characterization, Biological Evaluation, and In Silico Studies of Imidazolium-, Pyridinium-, and Ammonium-Based Ionic Liquids Containing n-Butyl Side Chains"

_molecules, 2022, doi:10.3390/molecules27196650_

Round 1
Reviewer 1 Report
The manuscript entitled " Synthesis, Characterization, Biological Evaluation, and Insilico Studies of Imidazolium, Pyridinium and Ammonium based Ionic Liquids Containing n-Butyl Side Chains" by Rabia Hassan and etal used the green chemistry synthesis procedure metathesis to develop three series of ionic liquids namely 1-methyl-3-butyl imidazolium, butyl pyridinium and diethyldibutyl ammonium cations whereas, Bromide (Br¯), Methanesulphonate (CH3SO3¯), Bis(trifluoromethanesulfonyl)imide (NTf2¯), Dichloroacetate (CHCl2CO2¯), Tetrafluoroborate (BF4¯), and Hydrogen sulphate (HSO4¯) as anions. The antibacterial potential was further investigated by insilico studies, and it was observed that Bis(trifluoromethanesulfonyl)imide (NTf2¯) containing imidazolium and pyridinium ionic liquids shows maximum inhibition against targeted bacterial strains and could be utilized in antibiotics.
This manuscript is valuable for the readers of Molecules after revision
I have the following comments on the manuscript
- - Revise the whole manuscript for typographical and grammar errors
- - Revise section 2.3. synthesis: use small letters
- - Redesign Schemes
- - Revise the number of NMR signals
- - 13CNMR spectra should be measured
- - Explain the biological activity
- Revise references
Author Response
Reviewer Report 1
The manuscript entitled " Synthesis, Characterization, Biological Evaluation, and Insilico Studies of Imidazolium, Pyridinium and Ammonium based Ionic Liquids Containing n-Butyl Side Chains" by Rabia Hassan and etal used the green chemistry synthesis procedure metathesis to develop three series of ionic liquids namely 1-methyl-3-butyl imidazolium, butyl pyridinium and diethyldibutyl ammonium cations whereas, Bromide (Br¯), Methanesulphonate (CH3SO3¯), Bis(trifluoromethanesulfonyl)imide (NTf2¯), Dichloroacetate (CHCl2CO2¯), Tetrafluoroborate (BF4¯), and Hydrogen sulphate (HSO4¯) as anions. The antibacterial potential was further investigated by insilico studies, and it was observed that Bis(trifluoromethanesulfonyl)imide (NTf2¯) containing imidazolium and pyridinium ionic liquids shows maximum inhibition against targeted bacterial strains and could be utilized in antibiotics.This manuscript is valuable for the readers of Molecules after revision
I have the following comments on the manuscript
- - Revise the whole manuscript for typographical and grammar errors
Response: Corrected
- - Revise section 2.3. synthesis: use small letters
Response: Corrected
- - Redesign Schemes
Response: Corrected
- - Revise the number of NMR signals
Response: Revised all the 1H and 13C NMR data in ascending order.
- - 13CNMR spectra should be measured
Response: 13C for butyl imidazole bromide, butyl pyridinium bromide has been added. Moreover, all other 13C were present but now we have written these in chronological order as suggested by reviewer.
- - Explain the biological activity
Response: biological activity has been explained in the in vitro testing as well as in silico results can better define the biological activity.
- Revise references
Response: Done
Reviewer 2 Report
The manuscript by Hassan et al. reports the results of the study of antibacterial activity of three types of ionic liquids with different counter anions. Despite the fact that the results are carefully presented, I cannot recommend the acceptance of the manuscript due to the following reasons:
1. The synthesis of previously known ILs is straightforward and follows the reported procedures. The completeness of the anion exchange reaction was not confirmed, thus the composition of each of 18 IL remains unknown.
2. The studied ILs are very common, thus a lot of research was already devoted to their toxicity, including towards bacteria. Even a quick search results in a number of papers devoted to the antibacterial effects of C4mim-type ionic liquids, e.g. 10.1016/j.ecoenv.2015.03.026, 10.5560/znb.2013-3150, 10.1039/C000899K
3. DFT and molecular docking calculations are used, but how the role of the counter anion was taken into account, is not discussed in the text, although that is the key difference of the studied ILs of one series.
Author Response
Reviewer Report 2
The manuscript by Hassan et al. reports the results of the study of antibacterial activity of three types of ionic liquids with different counter anions. Despite the fact that the results are carefully presented, I cannot recommend the acceptance of the manuscript due to the following reasons:
The synthesis of previously known ILs is straightforward and follows the reported procedures. The completeness of the anion exchange reaction was not confirmed, thus the composition of each of 18 IL remains unknown.
Response: The completeness of exchange reaction was confirmed with TLC as well as from the formation of halide salt which was precipitated out.
- The studied ILs are very common, thus a lot of research was already devoted to their toxicity, including towards bacteria. Even a quick search results in a number of papers devoted to the antibacterial effects of C4mim-type ionic liquids, e.g. 10.1016/j.ecoenv.2015.03.026, 10.5560/znb.2013-3150, 10.1039/C000899K
Response:
We are thankful to the reviewer for this lead. Which only describes one anion. And total 2 ionic liquids. Specifically related to acetone. “Toxicity of two imidazolium ionic liquids, [bmim][BF4] and [omim][BF4], to standard aquatic test organisms: Role of acetone in the induced toxicity”
Our paper has the novelty as it has six different kinds of anions. We have synthesized 18 new ILs and evaluated their invitro activity with DFT studies. Moreover 3 different kind of cations were also compared in this study. Such detailed study where three cations with six anion in different combinations are present has not been studied.
- DFT and molecular docking calculations are used, but how the role of the counter anion was taken into account, is not discussed in the text, although that is the key difference of the studied ILs of one series.
Response: We have discussed the anion role with a comparison of the anions and Counter ion effect in section 3.4.
Reviewer 3 Report
I have thoroughly enjoyed reading the article. ILs are gaining attention in recent comprehensive research fields due to its inherent potential to assist a biphasic system. 1. page 2, line 74-77 needs to be rewritten. 2. table1. please provide reaction details in the footnote 3. scheme 2 please remove parentheses from the salt names. 4. scheme 3, counter ion missing pyridinium sm. 5. no parentheses needed for compound numbering 6. figure 2, can authors include details of the number of antibacterial assay performed. the results are the average and is there significance observed. please use prism software to present this assay. 7. in-silico studies do not provide clear evidence of the role of butyl moiety. authors can think of adding any additional study or experiment to make the in-silico studies more important.Author Response
Reviewer Report 3
I have thoroughly enjoyed reading the article. ILs are gaining attention in recent comprehensive research fields due to its inherent potential to assist a biphasic system.
- page 2, line 74-77 needs to be rewritten.
Reply: Corrections made in the revised version.
- table1. please provide reaction details in the footnote
Reply: Reaction details provided as suggested.
- scheme 2 please remove parentheses from the salt names.
Reply: Changes done as suggested.
- scheme 3, counter ion missing pyridinium sm.
Reply: Corrected in Revised version.
- no parentheses needed for compound numbering
Reply: Changes done in Schemes as suggested.
- figure 2, can authors include details of the number of antibacterial assay performed. the results are the average and is there significance observed. please use prism software to present this assay.
Reply: Figure 2 is drawn is Prism software and experiments were done in triplicate and details added in revised version.
- in-silico studies do not provide clear evidence of the role of butyl moiety. authors can think of adding any additional study or experiment to make the in-silico studies more important.
Reply: We appreciate the reviewer’s comment. We have added some portion in section 3.4. We will extend this work in the next project.
Round 2
Reviewer 2 Report
The authors gave their comments to the critical points from the previous review and modified thier manuscript. The comments of other reviewers were also taken into account. However, some questions remain:
1. TLC method usually does not work well for highly polar and ionic compounds. In the experimental part, please give the detais of TLC analysis - type of plates used, eluent, Rf values for the obtained products.
2. The purity of the samples used for biological studies is important for obtaining reliable results. To confirm the completeness of anion exchange, elemental analysis should be given.
Author Response
- TLC method usually does not work well for highly polar and ionic compounds. In the experimental part, please give the detais of TLC analysis - type of plates used, eluent, Rf values for the obtained products.
Response: The purity of the samples used for biological studies is important for obtaining reliable results. To confirm the completeness of anion exchange, elemental analysis should be given. Rf values were in the range of 60-70 % for the starting bromides. Disappearance of Bromides was monitored using above said solvent system. Details have been added in the manuscript (Experimental Part).
- The purity of the samples used for biological studies is important for obtaining reliable results. To confirm the completeness of anion exchange, elemental analysis should be given.
Response: Thanks to the reviewer for valuable suggestion, however we don’t have
CHNS analyzer which can measure liquid samples.
Reviewer 3 Report
the manuscripts details an interesting study of ILs.
Author Response
Thanks to reviewer for comments.
Round 3
Reviewer 2 Report
The manuscript may be accepted.